# Multi-Level Support Technology and Application of Deep Roadway Surrounding Rock in the Suncun Coal Mine, China

**DOI:** 10.3390/ma15238665

**Published:** 2022-12-05

**Authors:** Hengbin Chu, Guoqing Li, Zhijun Liu, Xuesheng Liu, Yunhao Wu, Shenglong Yang

**Affiliations:** 1College of Energy and Mining Engineering, Shandong University of Science and Technology, Qingdao 266590, China; 2Xinwen Mining Group Co., Ltd., Xintai 271219, China; 3State Key Laboratory of Mining Disaster Prevention and Control Co-Founded by Shandong Province and the Ministry of Science and Technology, Qingdao 266590, China

**Keywords:** deep mining, mining roadway, surrounding rock, numerical analysis, multi-level support

## Abstract

To solve these problems of poor supporting effect and serious deformation and failure of surrounding rock of mining roadway under deep mining stress, a FLAC-3D numerical calculation model is established with −800 m level no. 2424 upper roadway in the Suncun Coal Mine as the background to compare the stress, deformation, and failure law of surrounding rock of mining roadway under once support and multi-level support with the same support strength. It is found that the multi-level support technology has obvious advantages in the surrounding rock of the horizontal roadway on the 2424 working face. From this, the key parameters of multi-level support are determined, and the field industrial test is carried out. The results show that the overall deformation of the surrounding rock is obviously reduced after multi-level support. The displacement of the two sides is reduced by about 40%, the displacement of the roof and floor is reduced by about 30%, and the plastic zone of the roadway is reduced by about 75%. The peak value of concentrated stress decreases from 98.7 MPa to 95.8 MPa, which decreases slightly. The integrity and stability of the surrounding rock are excellent, and the support effect is satisfactory. The research can provide reference and technical support for surrounding rock control of deep high-stress mining roadways.

## 1. Introduction

With the continuous mining of coal resources, shallow coal reserves begin to decline sharply, gradually transfer of underground mining to deep [1,2,3,4]. According to statistics, coal resources with a buried depth of more than 1000 m account for about 53% of China’s total coal reserves [5,6,7,8,9]. After coal mining into kilometer depth, due to large rock pressure, high temperature, complex stress environment, and other factors [10,11,12,13], the stress concentration of the roadway surrounding rock intensifies after rock excavation [14,15]. Under the action of deep high stress, the problems of surrounding rock failure, large deformation, difficult support, and obvious dynamic pressure are prominent [16,17,18,19]. The stability of surrounding rock becomes a problem restricting safe and efficient deep mining of coal resources [20,21,22,23].

Around the surrounding rock control of deep roadway, domestic and foreign experts have carried out a lot of theoretical and experimental research and put forward a variety of new support concepts and technologies. Kang et al. [24] put forward the collaborative control technology of support-modification-pressure relief for strong mining roadways in the soft rock of a kilometer deep well. Malan D.F et al. [25] considered that the support structure could better control the strain of the surrounding rock within the allowable range, and the deformation of the surrounding rock of the roadway can be reduced by adding more support structures. Wang et al. [26] put forward the high resistance yield pressure and high strength supporting technology of high strength bolt, strong anchor cable, and grouting reinforcement roadway and determined the supporting time of each supporting stage. Li et al. [27] put forward the support strategy of “bottom coal grouting reinforcement + high prestressed strong bolt anchor cable timely support”, which improved the strength of surrounding rock and support resistance of the mining roadway.

The current study provides strong technical support for the stability of surrounding rock in deep mining roadways. However, due to the characteristics of high stress, frequent dynamic load disturbance, and complex geological structure, it is often difficult to ensure the safety of surrounding rock with one support. Sometimes strong support can effectively control the deformation of surrounding rock, but the cost is very high. Therefore, multi-level support has been recognized by some scholars and field engineers. Liu et al. [28] adopted stepped combined support, advanced support, once support, and secondary support of the top of the heading face cooperated with each other, spray anchor grouting, and other means to realize the control of excavation construction safety and surrounding rock stability. Li et al. [29] analyzed the stress and strength adjustment process of surrounding rock in the support process of soft rock roadway and combined with the rheological mechanics model of soft rock roadway, deduced the theoretical formula of the optimal time of secondary support, and used it in engineering practice to better control the stability of roadway surrounding rock. Through practice, Li et al. [30] found that bolt length, initial anchoring force, stiffness, and other multi-level support parameters play a vital role in controlling roadway deformation and roadway safety.

However, at present, there is no better solution for how determining the support parameters in multi-level support technology. Based on the engineering background of the upper roadway of 2424 working faces at −800 m level in the Suncun Coal Mine, this paper proposes a graded support scheme through field measurement and numerical calculation. According to the deformation law of roadway surrounding rock and the design principle of roadway support, the support design is improved to ensure the safety and stability of the mining roadway in the working face.

## 2. Project Profile

The Suncun Coal Mine −800 m level 2424 working face roadway has a buried depth of about 1300 m, a length of about 1200 m, a trapezoidal section, a net width of 4.5 m, a left high of 2.3 m, a right high of 3.6 m. The right side of the roadway is the entity coal side. The thickness of the coal seam in the working face is 2.1~2.4 m, the average coal thickness is 2.3 m, and the average dip angle is 28°. The coal seam roof is silty sandstone, grayish black, stratified development, containing plant debris fossils, compressive strength of 21.4 MPa, and average thickness of 2.0 m; the coal seam floor is gray sandstone, stratified development, compressive strength is about 66.7 MPa, thickness 6~10 m. The mechanical parameters of the roof and floor footwall, and hanging wall are shown in Table 1.

## 3. Design of Support Parameters of Mining Roadway

### 3.1. Model Establishment

Based on actual conditions of no. 2424 upper roadway, considering the hosting of coal seam and mining mode, and considering the influence of boundary effect and advanced support pressure, make the model meet the full extraction [31]. The FLAC3D calculation model is established to simulate the stress distribution and deformation of surrounding rock after primary and tertiary support. The model size is 250 m × 300 m × 180 m and is divided into 2,358,781 elements, as shown in Figure 1. Using the Mohr–Coulomb model, the floor is fixed, the sides are horizontally simply supported, and the top is a free border. Owing to the roadway depth of 1300 m, the vertical stress of 29.05 MPa is applied to the upper surface of the model, and the horizontal displacement constraint is applied to the rest of the model.

### 3.2. Numerical Calculation Scheme and Results

To compare the surrounding rock control effect of different support methods, two simulation schemes of once support and tertiary support are set up under the same support strength, and the experimental results are analyzed and compared as follows. The supporting materials used are shown in Table 2.

#### 3.2.1. Scheme 1: Once Support

Immediately after the excavation of the roadway, no. 2424 upper roadway is supported, and bolt, anchor cable, and diamond mesh combined support are carried out in the whole roadway. Roof centered on the central axis of roadway, installation of Φ22–4100 mm mining steel strand grouting anchor on both sides, each anchor cable lengthens anchorage, anchor cable spacing 1050 × 1000 mm; MSGLD-600 equal strength thread rigid resin bolt is used in the middle of the two sides; MSGLD-335 equal strength thread rigid resin bolt is used in the top and seat angles, and the row spacing is 1000 × 1000 mm. The roadway support is shown in Figure 2, and the numerical calculation results are shown in Figure 3.

It can be seen from Figure 3 that the vertical stress concentration in entity coal side on the right side of the roadway and its vertical downward peak reaches 98.7 MPa. The peak stress is about 3.5 m from the right side of the roadway, which is easy to crush part of the coal body in the roadway; with once support, there is a certain degree of deformation in the roadway, and the maximum displacement of the roof is about 125 mm, the maximum deformation of the floor is about 50 mm, the maximum displacement of the two sides is 100 mm; compression-shear failure mainly occurs around the adjacent roadway of working face, but tension-compression failure occurs above the roof cutting line. The roof failure range of the plastic zone of the roadway surrounding rock is 0~20 m, and the plastic zone of the main roof of the roadway is basically bounded by the roof cutting line; the plastic yield zone is along the empty side, and the rest is the elastic zone.

Analysis shows maximum roof displacement near and above the central axis of the roadway, and the maximum displacement of surrounding rock appears in the middle of two sides. Therefore, the surrounding rock in the middle and upper part of the roof and the middle of the two sides is the key part of the support, and the bolt anchor should be used in time for support. Second, supporting should improve the support of the top angle and bottom angle on the basis of once support. The third advance support should be supported in the area with large deformation.

#### 3.2.2. Scheme 2: Tertiary Support

Support no. 2424 upper roadway immediately after excavation, application of mine steel strand grouting anchor support in roof central axis and it is upper position, length of anchor cable 4.1 m, anchor cable column interval 1050 mm × 1000 mm; The middle of the two sides adopts MSGLD-600 rigid resin bolt with equal strength thread. The row spacing between the bolts is 1000 mm × 1000 mm, and the length is 2.2 m.

After once support of the roadway reaches a new balance, secondary support for no. 2424 upper roadway, installation of mine grouting anchor at top angle of roof; the top and bottom angles of the two sides are reinforced by MSGLD-335 equal strength thread rigid resin bolting, and the spacing between the equal strength thread rigid resin bolt used in the once support is 1000 mm.

When the excavation working face is 50 m, the advance support is 200 m. Install grouting anchor cable in the area with large roof deformation to complete advanced grouting reinforcement.

According to the principle of graded support, the specific multi-level support method is shown in Figure 4. The numerical calculation results are shown in Figure 5.

It can be seen from Figure 5 that the vertical stress concentration occurs on the entity coal side of the right side of the roadway. The vertical stress peak is about 3.5 m from the right side of the roadway, and the size is 95.8 MPa. In the multi-level support, there is a certain degree of deformation in the roadway surrounding rock; the roadway roof subsidence is 75~100 mm, the maximum deformation of the floor is 25 mm, the maximum deformation of the two sides is 50 mm; The adjacent roadway around the working face is mainly a compression–shear failure, but there is tension-compression failure above the cutting line. The plastic zone of the main roof of the roadway is basically bounded by the top-cut line, and the failure range of the plastic zone of the roof is between 0~5 m. The plastic yield zone is along the goaf side, and the rest is the elastic zone.

### 3.3. Supporting Effect Comparison

Comparison of numerical simulation results of once support and tertiary support. From Figure 3a and Figure 5a, the peak stress concentration on the right side of the coal side is reduced from 98.7 MPa to 95.8 MPa, with a decrease of about 3%. From Figure 3b and Figure 5b, the maximum subsidence of the roof is reduced from 125 mm to 100 mm by 20%. The displacement of the two sides is greatly reduced from the maximum displacement of the once support 100 mm to 50 mm. Although the floor is not supported, the floor deformation is almost the same, with most of the location of the maximum deformation of 25 mm; From Figure 3c and Figure 5c, the failure range of the roof plastic zone is reduced from 0~20 m to 0~5 m with a reduction of about 75%. At the same time, the range of the plastic zone of the floor surrounding the rock of the multi-level support roadway reduces by about 20%, and the range of the plastic zone of two sides surrounding the rock is similar. The specific comparison effect is shown in Table 3.

In summary, after multi-level support, the stress environment of the roadway surrounding rock has been greatly improved. The roof subsidence and two sides are relatively stable, but there is still a certain degree of deformation. Through the comparison of the support effect, it is determined that the three-levels support scheme is adopted in the no. 2424 upper roadway of −800 m level in the Suncun Coal Mine.

## 4. Engineering Applications

### 4.1. Support Scheme Design

Based on the above research, the combined support of ‘grouting anchor cable + equal strength thread steel resin bolt + U-shaped steel connecting beam’ was used in no. 2424 upper roadway and the tertiary support scheme were adopted, construction scheme was consistent with the tertiary support simulation scheme. Once support is the basic support, that is, the immediate support after roadway excavation, secondary support is the reinforcement support of the top and bottom angle of the roadway when once support reaches a new balance. When the secondary support was completed, the working face pushed 50 m, and the advanced support was 200 m. According to the principle of graded support, the specific multi-level support method is shown in Figure 4.

### 4.2. Support Effect Monitoring

To verify the effect of the control technology proposed above in field application. During the tunneling period of the working face, a surrounding rock deformation monitoring station is set up every 40 m; the station follows a head-on setting. A total of 32 deformation monitoring stations were set up at different positions of no. 2424 upper roadway roof and floor. Using a laser range finder to monitor the displacement of the roadway roof and floor and two sides to obtain the total deformation of surrounding rock at each measuring point of the upper roadway. The monitoring results are shown in Figure 6.

It can be seen from Figure 6, after the no. 2424 upper roadway adopts three level supports, the maximum total deformation of each measuring point on the upper side was 50 mm; maximum deformation of the underlier was 40 mm; the maximum roof subsidence was 70 mm; floor because of no support, displacement was large, many positions of more than 100 mm. After on-site detection of the roadway roof and floor and two sides of the deformation within the allowable range, it can be seen that after the three-level support, the overall deformation of the roadway surrounding rock was small, and roadway deformation was more stable.

The deformation value of the roadway surrounding rock is analyzed by monitoring, and the error of numerical simulation results is calculated based on the monitoring results. The comparison results are shown in Table 4. It is found that the error of deformation value on both sides is less than 10%, which shows that the simulation effect is good. Because the floor is not supported, it is affected by other roadways during the construction, and the error is large. The roof subsidence error is about 30%, and the deformation value is less than the numerical simulation results. In summary, the deformation of the roadway surrounding rock is well controlled when multi-stage support is adopted. Numerical simulation results can provide some theoretical support for field application.

2424 roadway mining process, advance 200 m surrounding rock site photos are shown in Figure 7. The roof subsidence of the roadway was small, there was no large area of coal wall falling on both sides, the floor heave was small, and the surrounding rock integrity was good. Therefore, after the three-level support, the overall deformation of the surrounding rock of the roadway was well controlled.

## 5. Conclusions

(1)The numerical calculation models of once support and three-levels support of no. 2424 upper roadway in Suncun Coal Mine are established. Simulation results show that compared with once support, the deformation of surrounding rocks is obviously reduced in three-level support, in which the convergence of two sides reduces by about 40%, and the roof subsidence reduces by about 30%. The stress concentration of surrounding rocks also reduces, in which the concentrated stress of the coal side reduces by about 3%. The maximum plastic zone height in the roof strata decreases by about 75%.(2)The three-level support technology of ‘grouting anchor cable + equal strength thread rigid resin bolt + U-shaped steel connecting beam‘ is determined to support no. 2424 upper roadway, which includes primary base support, secondary reinforcement support, and advanced tertiary support. Field monitoring shows that the maximum roof subsidence is about 70 mm, the side-to-side convergence is less than 50 mm, and the floor heave deformation is less than 25 mm.(3)Compared with once support technology, the multi-level support technology on the condition with the same total support strength can reduce the roadway deformation and stress concentration of two sides to a certain extent to improve the roadway integrity and stability.

## Figures and Tables

**Figure 1 materials-15-08665-f001:**
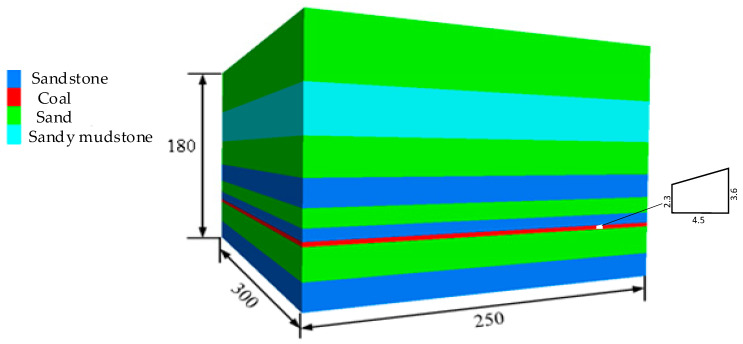
Diagram of the numerical model.

**Figure 2 materials-15-08665-f002:**
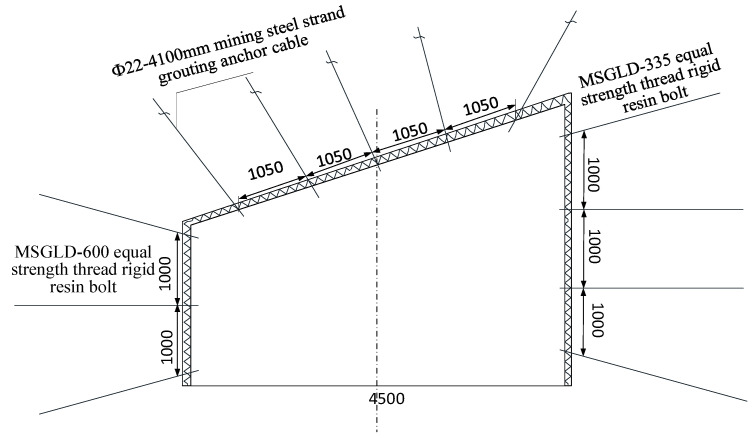
Roadway once support section diagram.

**Figure 3 materials-15-08665-f003:**
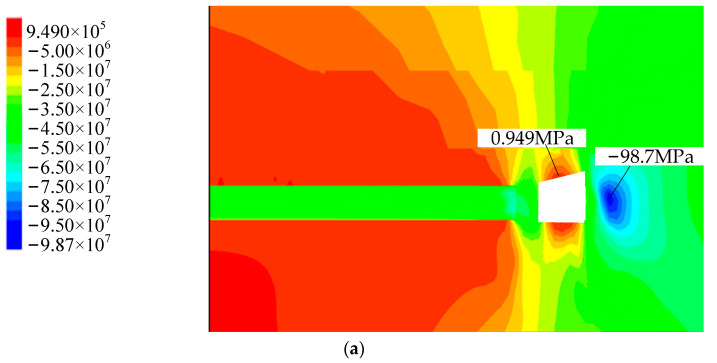
Simulation results of once support, (**a**) vertical stress distribution diagram, (**b**) vertical displacement distribution map, (**c**) distribution diagram of the plastic zone.

**Figure 4 materials-15-08665-f004:**
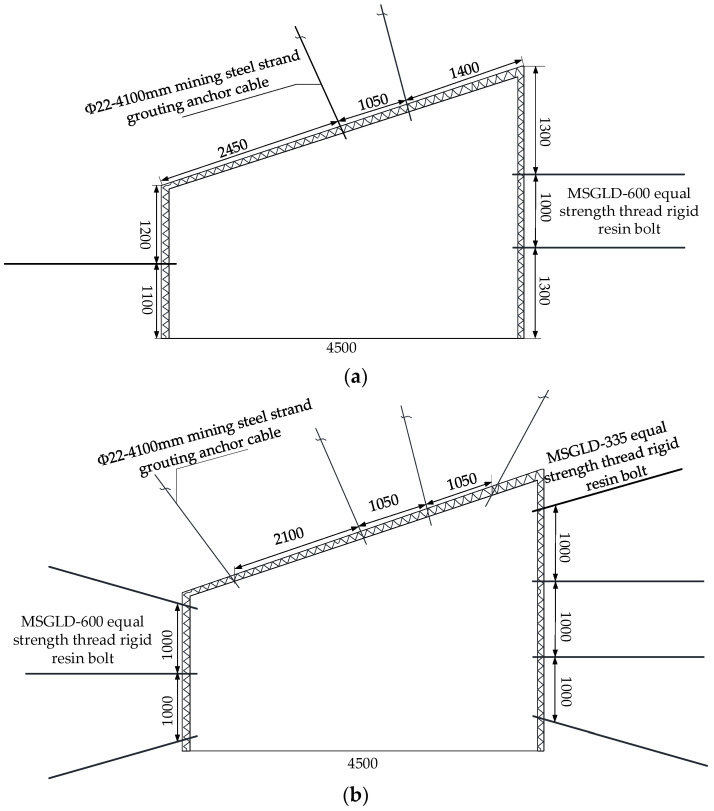
Tertiary support sectional drawing: (**a**) primary support, (**b**) secondary support, (**c**) tertiary support.

**Figure 5 materials-15-08665-f005:**
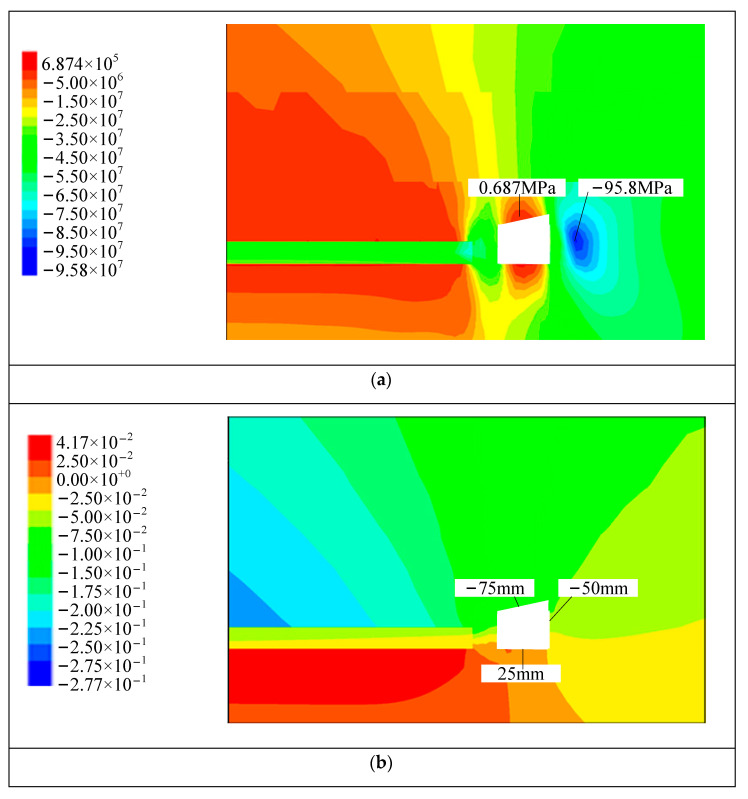
Tertiary support simulation results, (**a**) vertical stress distribution diagram, (**b**) vertical displacement distribution map, (**c**) distribution diagram of the plastic zone.

**Figure 6 materials-15-08665-f006:**
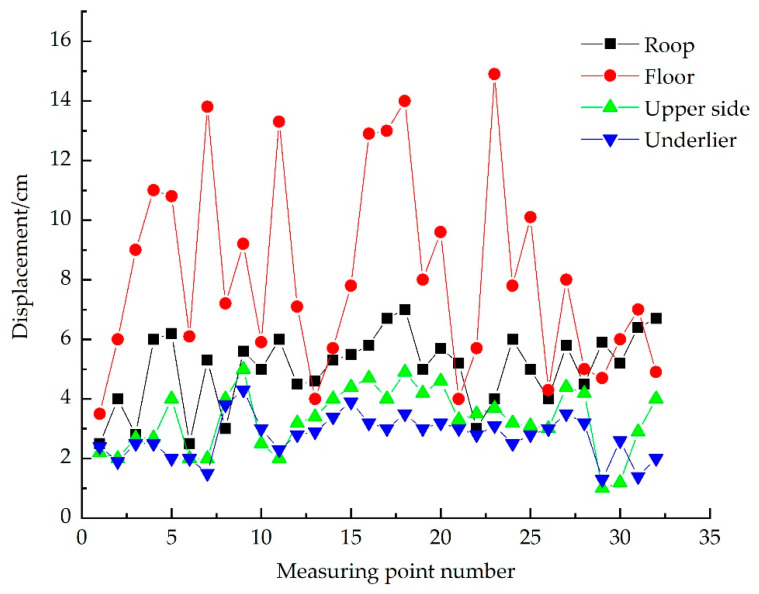
Total deformation of surrounding rock.

**Figure 7 materials-15-08665-f007:**
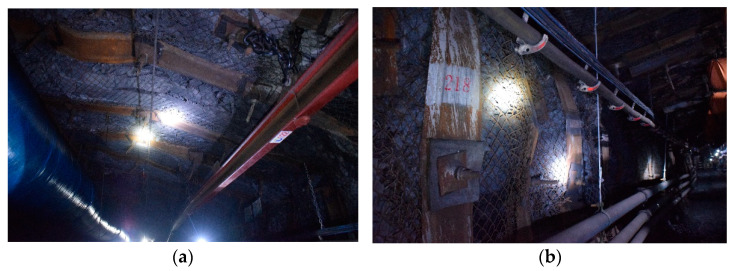
Surrounding rock control effect diagram: (**a**) roof, (**b**) upper side.

**Table 1 materials-15-08665-t001:** Mechanical parameters of the roof and floor footwall and hanging wall.

Rock Name	Thickness/m	Volumetric Weight/(kN/m³)	Bulk Modulus/GPa	Shear Modulus/GPa	Internal Friction Angle/°	Tensile Strength/MPa	Cohesion/MPa
Sandstone	6.0~10.0	25	11.5	7.3	28	8.4	2.6
Siltstone	0~4.0	23	2.1	1.8	32	4.3	0.8
4#coal	2.1~2.4	14	2.1	0.93	24	0.7	0.5
Sandstone	5.0~6.0	25	11.5	7.3	28	8.4	2.6

**Table 2 materials-15-08665-t002:** Properties of supporting materials.

Support Material	Size	Yield Strength/MPa	Rod Elongation	Elastic Modulus/MPa	Poisson Ratio
mining steel strand grouting anchor cable	Φ22 × 4100 mm	≥1860	≥2%	195 × 10^3^	0.3
MSGLD-600 equal strength thread rigid resin bolt	Φ22 × 2200 mm	≥600	≥15%	206 × 10^3^	0.3
MSGLD-335 equal strength thread rigid resin bolt	Φ20 × 2000 mm	≥350	≥15%	200 × 10^3^	0.3

**Table 3 materials-15-08665-t003:** Comparison of relevant parameters of different schemes.

Scheme	Maximum Stress of Coal Side/MPa	Roof Subsidence/mm	Two Sides Displacement/ mm	Roof Plastic Zone Range/m	Floor Deformation/m
Once support	98.7	75~125	50~100	0~20	25~50
Tertiary support	95.8	50~75	25~50	0~5	25

**Table 4 materials-15-08665-t004:** Comparison of numerical and analytical results.

Project	Average Roof Subsidence/mm	Average Floor Heave/mm	Average Upper Side Displacement/mm	Average Underside Displacement/mm
Numerical simulation	65	30	30	30
Analytical results	50.2	81.3	33.1	27.6
Error	29.5%	63.1%	9.4%	8.7%

## Data Availability

Not applicable.

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
