# Peer review of "Multi-Level Support Technology and Application of Deep Roadway Surrounding Rock in the Suncun Coal Mine, China"

_materials, 2022, doi:10.3390/ma15238665_

Round 1

Reviewer 1 Report

Solving the problem of stability of mining operations is one of the most important issues in mining. The paper is current, well presented and with applicable results. Using the FLAC 3D program leads to good results that can lead to the identification of suitable support systems.

Author Response

Thank you for your comments on our manuscript.

Best regards.

Reviewer 2 Report

When I reviewed and evaluated the article with the title “Multi-level support technology and application of 2424 Upper-level roadway surrounding rock at -800m level in Suncun Coal Mine”, the deficiencies stated below in the article should be eliminated. If these major deficiencies are eliminated, it is appropriate for me to publish the article in "Materials".

1. The English of the paper is poor. Required to be checked by a native English speaker.

2. Numerical and analytical results should be given as tables and the results should be compared.

3. Line 81, Footwall and hanging wall can be used instead of coal rock

4. Adding the scheme of the project work area can make the paper more understandable.

5. Vertical stress of 29.05 MPa (Why?)

6. Figure1. Roadway section can be shown on the figure.

7. The support material used might be given as a table with its properties.

8. Tertiary support diagram can be given before numeric (FLAC-3D) figures.

9. In general, the paper should be written clearly and rearranged. Numerical studies can be given after analytical studies.

Best regards

Author Response

Thanks for your valuable advice, which not only gives us a clear understanding of the shortcomings and improvements of the paper but also points out the direction for the revision.

Responses to the comments from the reviewers. Please see attachment.

Author Response

(The authors gave the same response as above.)
